# Capsule-Based Self-Healing and Self-Sensing Composites with Enhanced Mechanical and Electrical Restoration

**DOI:** 10.3390/polym14235264

**Published:** 2022-12-02

**Authors:** Georgios Foteinidis, Maria Kosarli, Pantelis Nikiphorides, Kyriaki Tsirka, Alkiviadis S. Paipetis

**Affiliations:** Department of Materials Science and Engineering, University of Ioannina, 45110 Ioannina, Greece

**Keywords:** composites, self-sensing, self-healing, nanomodified, nanocomposites, smart materials, SHM, electrical restoration

## Abstract

In this work, we report for the first time the manufacturing and characterization of smart multifunctional, capsule-based self-healing and self-sensing composites. In detail, neat and nanomodified UF microcapsules were synthesized and incorporated into composites with a nanomodified epoxy matrix for the restoration of the mechanical and electrical properties. The electrical properties were evaluated with the use of the impedance spectroscopy method. The self-healing composites were subjected to mode-II fracture toughness tests. Additionally, the lap strap geometry that can simulate the mechanical behavior of a stiffened panel was used. The introduction of the nanomodified self-healing system improved the initial mechanical properties in the mode-II fracture toughness by +29%, while the values after the healing process exceeded the initial one. At lap strap geometry, the incorporation of the self-healing system did not affect the initial mechanical properties that were fully recovered after the healing process.

## 1. Introduction

With the increasing use of composites in the aerospace market, it was necessary to develop new and advanced, multifunctional composite systems with an inherent capacity to detect or sense any mechanical damage or chemical degradation as well as the ability to heal. A non-destructive inspection is a typical procedure which requires human intervention, and its application is limited to the dimensions of the structure and takes place only periodically [1,2]. 

The structural health monitoring (SHM) approach can provide useful information about the structural integrity of the material. A complete system should include sensors or actuators, data acquisition systems, power storage devices and an appropriate software for data analysis [1,2]. The sensors used for the SHM are based on a non-destructive evaluation (NDE). With this approach, the periodic inspections of a component can be replaced. The user or the inspector can be informed immediately in the case of any damage (online structural health monitoring) from the sensors incorporated inside the material or attached externally [3]. A wide variety of sensors can be used for the SHM of a structure, including conventional resistance strain gauges [4], fiber optic sensors [5], piezoelectric wafer active sensors [6], etc. Each type of sensor exploits different physical principles of the material in order to monitor its health [7]. Bekas et al. developed a planar interdigital capacitive sensor that was attached (by printing) to the surface of a composite joint in order to monitor the curing process of the adhesive and the progress of developed damage [8].

However, the use of external sensors either on top or inside the composite can degrade the structural performance since the sensors may act as damage initiation sites or internal defects [9,10]. In this case, many researchers have focused on the development of smart materials, and led to the production of structures with various functionalities, such as self-sensing [11], self-healing [12,13], electromagnetic shielding [13], energy harvesting [14,15], etc. Self-sensing is a valuable capability since the composite can identify internal deformations [16,17] and in-service damages [11,18], as well as additional functionalities such as curing monitoring [19] without the usage of external or embedded sensors. 

Self-sensing composites are based on the exploitation of the inherent properties of their phases [20,21]. By definition, composites consist of two (or more) phases with different properties [22]. Extensive research has been based on the electrical conductivity of the carbon fibers and their unique electrical anisotropy to detect strain and damage or identify the damage location [17,23,24]. Another approach in recent years has been the addition of a third nanophase in non-conductive composites, i.e., glass fiber-reinforced composites (GFRPs) [11,18]. One of the most promising nanophases is carbon-based nanomaterials, i.e., carbon nanotubes (CNTs), graphene, carbon black, etc. [11,25,26]. These nanomaterials pertain to the composite’s additional properties as an improved electrical conductivity, apart from the enhancement of the mechanical properties. A fundamental parameter for the attainment of the electrical conductivity is for the concentration of the nanophase to exceed a specific value, the percolation threshold [27,28,29]. Above this concentration, the nanophase can form a continuous network over the volume of the composite that imparts electrical conductivity. This conductive network of the nanophase can act as a sensor that is capable of detecting changes in the structure of the composite with a high sensitivity. 

Another functionality of polymeric composite materials is the self-healing functionality [30]. Self-healing materials are state-of-the-art technologies capable of healing and restoring internal damage or functionalities in a material [31]. Inspired by nature, these materials can repair themselves, even without an external intervention. The concept of self-healing composites can be classified as intrinsic or extrinsic according to the mechanism with which the healing agent is sequestered into the material [32]. The intrinsic self-healing principle is that polymers can restore their initial properties through chemical or physical processes such as ionomeric bonding, thermally reversible reactions, or molecular diffusion [33,34]. Extrinsic self-healing is a process that involves the delivery of healing components to a targeted site in a material, and there is no need to modify the polymer [35,36]. 

The capsule-based method focuses on encapsulating the healing agent into discrete capsules at micro or nano dimensions and their integration into the material [37]. The healing process starts with the rupture of the polymeric capsule shell-wall [38,39]. Although, there is only one local healing cycle since, after damage, the healing agent is consumed (locally) [40]. Tsilimigra et al. reported a low-content capsule-based self-healing carbon reinforced composite manufactured by the wet layup method [41]. The UF microcapsules loading was only at 10 wt.% while Scandium (III) Triflate (Sc(OTf)_3_) was used as a catalyst. The composites were subjected to three-point-bending and mode-II fracture toughness tests, while reference specimens were manufactured to evaluate the knockdown effect. A significant recovery of 84% in the fracture toughness was reported in terms of the healing efficiency.

Besides all the advantages of self-healing materials, the traditional self-healing composites are not capable of restoring the conductive network of the nanophase after the healing of the damage, but they have limited only the restoration of the mechanical properties. As a result, the self-sensing functionality is terminated due to a network disruption. In this research, self-sensing and self-healing multifunctional GFRPs were manufactured, capable of restoring both functionalities simultaneously. A nano-enhanced epoxy matrix with dispersed multiwall CNTs and CB was utilized as the sensitive element. The self-healing properties were achieved with the incorporation of nanomodified microcapsules for the restoration of both mechanical and electrical properties. The composites were tested under a mode-II Interlaminar fracture toughness configuration while the new proposed lap strap geometry [17,42] was also used. Impedance spectroscopy scans were applied to the specimens before and after the damage and healing processes, in order to examine the restoration of the electrical properties. Additionally, for the first time, IS measurements were performed online during the damage and healing processes of the composites for the demonstration of the self-sensing functionality. 

## 2. Materials and Methods

### 2.1. Materials

A two-part low viscosity epoxy resin Araldite LY5052 and hardener Aradur CH 5052, provided by Huntsman Advanced Materials, Basel, Switzerland, at a mix ratio of 100:38, was used as the matrix phase. The epoxy matrix was nanomodified using the Graphistrength C-100 MWCNTs supplied by ARKEMA, Colombes Cedex, France and Ketjenblack carbon black from Nanocyl S.A with a mean agglomeration size of 5–10 μm. As a primary reinforcement, the unidirectional (UD) glass fabric with an areal density of 320 g/m^2^ and the biaxial (BI) glass fabric at ±45° with an areal density of 240 g/m^2^ supplied by Fibermax S.A were also used.

The diglycidyl ether of bisphenol-A (DGEBA, Epikote 828 lvel) epoxy resin, by Dichem Polymers Greece, was chosen as a healing agent. The nanomodification of the DGEBA resin was accomplished using the same MWCNTs. The viscosity of the nanomodified healing agent was decreased using a non-toxic solvent, ethylphenylacetate (EPA), at 15% wt. Urea (NH_2_CONH_2_) and formalin (37% wt. in H_2_O) were used as the main wall-forming materials. Ammonium chloride (NH_4_Cl) and resorcinol (C_6_H_4_-1,3-(OH)_2_) were used as the stabilizers. The poly (ethylene-maleic-anhydride) (EMA, Mw = 100.000–500.000 g/mol) copolymer powder was used as the surfactant. The incorporated catalyst was the Aluminum (III) triflate (Al(OTf)_3_) purchased by Sigma-Aldrich, St. Louis, MO, USA.

### 2.2. Encapsulation Process

The microcapsules were manufactured according to our previous studies via the in situ emulsification polymerization method [12,13,37]. Two types of microcapsules were produced for the tests, neat (with a neat healing agent) and nanomodified (with a nanomodified healing agent). The stirring rate was chosen to be 500 rpm. A dilution percentage of 15 wt.% was used, and the reaction was left to proceed for 4 h. When the encapsulation process ended, the mixture was left to cool down at room temperature, and the suspended microcapsules were rinsed with ethanol using a Buchner filter. The capsules were left to dry in a laboratory oven under ambient conditions. Hereafter, the produced capsules will be referred to as “neat” for capsules containing the unmodified (neat) healing agent and “nanomodified” for capsules containing the nanomodified healing agent (with 0.5 wt.% MWCTNs).

### 2.3. Manufacturing of Mode-II Specimens

Three composite laminates were manufactured for each scenario (reference, neat and nanomodified systems) using 16 plies of the UD glass fabric. The lamina thickness was approximately 0.26 mm. The manufacturing process was the hand lay-up technique with an applied pressure of 3 MPa, while curing took place at 25 °C for 24 h. Spacers were used in order to ensure a uniform laminate thickness of 4.22 mm. As the matrix phase, the nanomodified LY 5052 epoxy resin was used. The self-healing system (neat or nanomodified) was placed at the midplane of the laminate. The healing system contained 25% wt. capsules and 2.5% wt. Aluminium (III) triflate (Al(OTf)_3_). Capsules and catalyst were dispersed into the epoxy (at a different beaker) and then applied in the middle of the composite. A thin high-temperature release film was also placed in the mid-thickness plane to act as an initial pre-crack. After curing, the specimens were cut at the desired dimensions according to AITM-1.0006 [43]. At each specimen, carbon fabric squares with dimensions of 20 × 20 mm were attached using the matrix material on both sides of the specimen to form electrodes, as depicted in Figure 1. In addition, at each carbon patch, one cable was also attached using conductive silver paste. The patches were applied using a vacuum bagging process and left for curing for 24 h at room temperature. Hereafter, the produced specimens will be referred to as the “reference” for the unmodified specimens (no self-healing system), as the “neat system” for the specimens containing neat capsules, and the “nanomodified system” for specimens containing nanomodified capsules.

### 2.4. Lap Strap Specimens

The lap strap specimen is a geometry that can simulate the stiffening of a composite panel. This geometry comprises two parts, the lap and the strap bonded with nano-reinforced resin as an adhesive layer between them. Both composite parts were extracted from a quasi-isotropic panel with lamination [±45/0/90] S, according to CWA 17896:2022. The dimensions and the geometry are depicted in Figure 2 (up). The lap strap specimens were manufactured by a hand lay-up technique and vacuum bagging. The self-healing system was incorporated only in the measured bonding area between the lap and the strap (Figure 2 down). For the modification of the adhesive, 25% wt. of the capsules and 2.5% wt. of the catalyst were dispersed into the matrix phase. The same concept with the mode-II specimens was followed here again. Three different types of specimens were manufactured: (i) without micro-capsules, (ii) with micro-capsules with a neat healing agent and (iii) with micro-capsules with a nano-enhanced healing agent. Hereafter, the produced specimens will be referred to as the “reference” for unmodified specimens (no self-healing system), as the “neat system” for specimens containing neat capsules and the “nanomodified system” for specimens containing nanomodified capsules.

In order to monitor the structural health and the restoration of the conductive network to the lap strap specimens, impedance spectroscopy was utilized simultaneously with the mechanical testing and the self-healing process. Figure 3 shows the lap strap geometry and the spots where the electrodes were attached. Two carbon fabrics, with dimensions of 20 × 10 mm and connected cables, were used as the electrodes, and they were attached with nano-reinforced resin on the lap and the strap. The failure of the specific geometry should be denoted by an increase in the impedance due to the disruption of the conductive path between the lap and the strap. While the adhesive between these two parts is debonding, the contact area of the two parts is reduced, and the conductive path length increases.

### 2.5. Mode-II Interlaminar Fracture Toughness

Mode-II tests were performed using a WDW-100 Jinan universal testing machine under three-point bending equipped with a 100 kN loadcell. The fracture of the interface occurring during this test is an in-plane shear fracture. The displacement rate was set at 1 mm/min while the load and cross-head displacement were recorded. According to [43], the critical mode-II strain energy release rate, G_IIc_, was calculated using Equation (1):G_IIc_ = [9  P a^2^ d  1000]/[2  w  (1/4  L^3^ + 3a^3^)](1)
where G_IIc_ (J/mm^2^) is the mode-II fracture toughness energy, P is the critical load, d is the displacement at the onset of the delamination, a is the initial crack length, w is the width of the specimen and L is the span length. The mechanical testing was stopped when the delamination at the mid-plane of the specimen occurred.

### 2.6. Lap Strap Specimen Testing Protocol—Modelled Structure Composites

Lap strap tests were performed under tensile mode at a WDW-100 Jinan universal testing machine equipped with a 100 kN load cell, according to CWA 17896:2022. The displacement rate was selected at 1 mm/min. The specimens were gripped 50 mm at either end, leaving an initial grip-to-grip separation of 100 mm. The stress was calculated as the ratio of the load to the cross-section area (thickness × width) and the strain as the ratio of the extension to the grip-to-grip distance. The stress and strain were recorded continuously during the test. All the tests were stopped upon the delamination of the strap from the lap.

### 2.7. Healing Process/Healing Efficiency of Mode-II Tests

The healing process was achieved in a laboratory oven at 80 °C for 48 h with an applied weight on the specimens. At the end of the healing process, the specimens were left to cool down at room temperature and retested under the same conditions. The healing efficiency was evaluated in terms of the peak load (Pc) and mode-II interlaminar fracture toughness energy (G_IIc_) recovery from Equations (2) and (3), respectively: n_p_ = P_c_^h^/P_c_^v^
(2)
n_G_ = G_IIc_^h^/G_IIc_^v^
(3)
where P_c_^h^ and P_c_^v^ are the peak load of the healed and the virgin specimens and the G_IIc_^h^ and G_IIc_^v^ represents the mode-II interlaminar fracture toughness energies of the healed and virgin specimens. The change in the initial mechanical properties after the self-healing system incorporation or the knockdown effect was evaluated using Equations (4) and (5).
k_p_ = 1 − (P_c_^v^/P_c_^r^)(4)
k_G_ = 1 − (G_IIc_^v^/G_IIc_^r^)(5)

### 2.8. Healing Process/Healing Efficiency at Lap Strap Specimens

The healing protocol that was followed to restore the mechanical and electrical properties of the lap strap specimens was the one described in the previous section (mode-II specimens). The healing efficiency was related to the efficiency in the recovery of the initial strength n_σ_ and according to Equation (6): n_σ_ = σ^h^/σ^v^
(6)
where σ^ν^ and σ^h^ correspond to the maximum stress at which the delamination initiated at the tip of the strap of the virgin and healed specimens, respectively. The change in the initial properties or the knockdown effect was evaluated using Equation (7).
k_σ_ = 1 − (σ^v^/σ^r^) (7)

### 2.9. Scanning Electron Microscopy

The mean diameter of the capsules was estimated from 100 different individual capsule measurements via SEM images using a JEOL JSM 6510LV, Oxford Instruments, Abingdon, UK. The microcapsules were initially gold-palladium sputter-coated, while the operating voltage was set at 5 keV.

### 2.10. Impedance Spectroscopy Measurements

Impedance spectroscopy measurements were accomplished using an Advanced Dielectric Thermal Analysis System (DETA-SCOPE) supplied by ADVISE, Chios, Greece. 

Impedance spectroscopy was used to assess the electrical properties of the specimens offline at two different stages, initially, before the damage and after the healing process. The scans were performed between 0.01 Hz and 100 kHz. The cables were connected directly with the attached cables of the specimens. A sinusoidal voltage of 10 V was applied between the two cables for the impedance measurements. The recovery of the electrical properties was estimated from the ratio of the DC conductivity values of the healed specimens (σ^Healed^) to the DC conductivity values of the initial (σ^Initial^) specimens through Equation (8).
e = σ^h^/σ^v^(8)

Additionally, impedance spectroscopy was exploited for the implementation of the online measurements. The online measurements offered an overview of the changes in the electrical properties of the specimens during the mechanical test and the healing process.

## 3. Results

### 3.1. SEM Images

Figure 4 shows the two SEM images of the produced nanomodified microcapsules that were incorporated at the composites obtained at two magnifications (i.e., ×330 and ×1000). The SEM images revealed that the capsules were spherical in shape and had a rough exterior shell wall. The rough, porous morphology of the outer surface can be easily observed at a higher magnification image. This is an important characteristic of the interlocking between the capsules and the matrix. The neat capsules had a mean size of 205.81 ± 26.32 μm. The mean diameter of the nanomodified capsules was calculated at 167.25 ± 10.24 μm from datasets of 100 measurements. 

### 3.2. Recovery of the Mode-II Fracture Toughness Energy

Figure 5 depicts the representative load–displacement plots of reference (neat system) and nanomodified system of mode-II specimens. It is observed that the load curves revealed a linear increase followed by a change in the slope from this linearity. When the crack propagated inside the specimen, a load drop was observed. This was a typical behavior observed in brittle epoxy composites [44]. Moreover, this trend was also observed in initial and healed specimens (Figure 6). After incorporating neat capsules, the G_IIc_ values increased by 27.98%, while in the case of nanomodified capsules, the values increased by 29.17%. However, the max load decreased with the addition of neat capsules by −6.80% and −2.83% in the nanomodified capsule system. This decrease can be attributed to the embedded capsules and catalyst that may induce the toughness through two mechanisms, according to [45]. The first mechanism corresponds to the thicker resin-rich regions that led to a higher plastic deformation. In contrast, the second one is based on the ability of the microcapsules to restrain the hackle formation [46].

Table 1 and Table 2 depict the calculated healing efficiency values in terms of the max load and G_IIc_. In neat capsule systems, the mode-II fracture toughness energy was recovered by 180.93%, while the max load was regained by 109.42%. In the case of the nanomodified capsule systems, the G_IIc_ was restored by 191.24% and the load by 117.28%. The nanomodified system recovered its properties ca. 10% more than the neat system due to the modification of the healing agent and the excellent properties of the CNTs. 

It should be mentioned that the values obtained after the healing process in neat and nanomodified systems in terms of the mode-II fracture toughness energy and in the maximum load exceeded the initial ones, resulting in high healing efficiencies above 100%. This phenomenon can be explained by the fact that the healing process is a thermal process similar to the post-curing process. Post-curing usually induces the toughness of the matrix and leads to higher mode-II fracture toughness properties [46]. In addition, voids that may be induced inside the material during the manufacturing process were eliminated with the healing process.

### 3.3. Recovery of the Electrical Properties on Mode-II—Offline

The electrical recovery was also examined in mode-II specimens via impedance spectroscopy. Regarding the mode-II specimens with neat capsules, the initial impedance value at 0.01 Hz was 1.15 × 10⁶ Ohm (black curve with filled squares), and the Ohmic to non-Ohmic transmission frequency was 1914.10 Hz (Figure 7). After the mechanical test and the self-healing process, the impedance at 0.01 Hz increased to 2.44 × 10⁶ Ohm (black curve with unfilled squares), and the critical frequency shifted to 920.77 Hz. The mean DC conductivity recovery was calculated at 61.21 ± 5.51%.

The mode-II specimens with incorporated capsules with a nano-enhanced healing agent which exhibited an improved electrical recovery after the self-healing process, as depicted in the representative plot in Figure 8. The magnitude of the impedance of this specimen, which was 2.66 × 10⁶ Ohm at 0.01 Hz (black curve with filled squares), was recovered at 3.10 × 10⁶ Ohm (black curve with unfilled squares). The Ohmic to non-Ohmic transition frequency (i.e., the transition from the rage where impedance is independent of frequency to the range that impedance depends on the frequency) was altered from 498.80 Hz to 569.10 Hz, which is a negligible change. The mean recovery value of the DC conductivity was 79.32 ± 4.41%. 

### 3.4. Recovery of the Electrical Properties on Mode-II—Online

The damage and self-healing processes were monitored online with impedance spectroscopy. The cables of the analyzer remained attached to the specimen during the procedure. The results of this experiment are presented in Figure 9. The online monitoring was accomplished in both categories of the mode-II specimens, in contrast with the lap strap specimens, in which the online was not feasible in specimens with a neat healing agent after the corruption of the CNT network, i.e., the mechanical test. An R-C in parallel equivalent circuit was selected to simulate the electrical behavior of the structure. Figure 9 (left) illustrates the mode-II specimen with neat capsules. The initial resistance was 1.65 × 10⁶ Ohm. At the second stage of the procedure, the damage stage, the resistance increased to 2.80 × 10⁶ Ohm. After the damage, the neat healing agent flowed out of the capsules, and the resistance was stabilized at 2.00 × 10⁶ Ohm at the end of the healing process. The mode-II specimen with a nano-enhanced healing agent had an initial resistance at 2.75 × 10⁶ Ohm (Figure 9, right). The resistance was increased at 3.77 × 10⁶ Ohm due to the damage of the specimen during the mechanical test. The final resistance after the healing process was 3.16 × 10⁶ Ohm. Both specimens exhibited a sufficient capability of monitoring the damage and the self-healing process, while the specimen with a nano-enhanced healing agent presented an improvement considering the electrical recovery. It should be noted that the recovery values of the specimens that were used for online monitoring were smaller than those of the specimens that were measured offline. This phenomenon occurred due to the difficulties of employing the measurement simultaneously with the application of heat and pressure. As a consequence of the cables that remained attached during the self-healing process, less pressure was applied to the specimen, resulting in reduced recovery values compared with the previous specimens. These specimens were excluded from the electrical and mechanical recovery results.

### 3.5. Recovery of Lap Strap Strength

Figure 10 depicts representative stress–strain curves from the reference, neat capsules system and nanomodified capsule system of lap strap specimens. A sudden and brittle failure of the adhesive between the lap and the strap occurred in all cases, which manifested in a slight drop at the stress–strain curve. The same mechanical behavior was observed after the healing process (healed specimens), as shown in Figure 11. Incorporating the healing system in both systems (neat and nanomodified) within the adhesive polymer did not affect the initial properties of the coupons. A negligible reduction in the initial strength properties of −2.07% was observed in neat systems since, in nanomodified systems, the same values were exhibited with the maximum reference stress. 

The successful recovery of the stress at the delamination point of the strap was also estimated (Table 3). In detail, neat systems restored their mechanical properties by 90.67% (from 228.08 ± 17.24 to 207.33 ± 39.53 MPa), while nanomodified systems increased their healed values from 233.68 ± 18.25 to 240.07 ± 32.21, resulting in a healing efficiency of 107.36%. As in the case of the mode-ii tests, the addition of the MWCNTs into the healing agent sufficiently improved the healing efficiency by ca. 17% compared to the neat capsule system.

### 3.6. Recovery of the Electrical Properties on Lap Strap—Offline

The concurrent recovery of mechanical and electrical properties in nanomodified capsule-based self-healing epoxies research included two types of lap strap specimens, with neat and with a nanomodified healing agent inside the capsules.

The impedance measurements on the lap strap specimens with neat capsules revealed that the initial magnitude of the impedance at 0.01 Hz was 10⁶ Ohm (Figure 12: black curve with filled squares). The Ohmic to Non-Ohmic transition appeared at 1857.00 Hz. After the damage and self-healing process, the impedance at 0.01 Hz increased to 5.77 × 10⁶ Ohm (Figure 12: black curve with unfilled squares), and the Ohmic to Non-Ohmic transition shifted to 1050.00 Hz. These changes declared that the CNT network was not regenerated, as the electric profile did not exhibit a restoration. The mean DC conductivity recovery was at 24.53 ± 5.48%.

Figure 13 illustrates the impedance measurements of the lap strap specimens with a nano-enhanced healing agent inside the micro-capsules. There was a slight increase in the impedance magnitude after the damage and self-healing process, i.e., the magnitude was increased from 1.37 × 10⁶ Ohm to 1.65 × 10⁶ Ohm at 0.01 Hz. The Ohmic to Non-Ohmic transition shifted from 2284.00 Hz to 1824.00 Hz. The mean DC conductivity recovered at 76.63 ± 8.04%.

### 3.7. Recovery of the Electrical Properties on Lap Strap—Online

The impedance measurement was applied online during the damage and self-healing process. Figure 14 illustrates the resistance of the R-C in a parallel equivalent circuit versus time. As shown in Figure 14 (up), the online monitoring of the lap strap specimens with neat capsules was unsuccessful due to a neat resin flow when the capsule wall corrupted. The damage to the lap strap can be distinguished within the first 10 min. After the damage stage, the neat resin of the capsules added an isolated phase between the lap and the strap, preventing the charge transfer between these components. As a result, the recorded data by the impedance analyzer were inaccurate. Contrariwise, the specimens with a nano-enhanced healing agent presented adequate damage and a self-healing monitoring capability. A resistance increase occurred, from 7.69 × 10^5^ Ohm to 6.96 × 10^6^ Ohm, due to debonding between the lap and the strap (Figure 14 (down)). Thereinafter, the resistance was decreased as the healing agent was flown out of the raptured capsules. The reaction of the healing agent with the catalyst continued until the end of the healing process.

It should be noted that the recovery values of the specimens used for online monitoring were smaller than those measured offline. As in the case of mode-II, this phenomenon occurred due to the difficulties of employing the measurement simultaneously with the application of heat and pressure. These specimens were excluded from the electrical and mechanical recovery results.

### 3.8. Fractography

The presence of the self-healing system inside the composite specimens and the rupture of the capsules after the mechanical testing of the specimens were confirmed by a fractographic SEM study. More specifically, SEM images were taken from the virgin and healed specimens. Figure 15 (left) illustrates an image from the virgin specimen which proves the presence of intact capsules and catalyst inside the composite. Figure 15 (right) illustrates an area from a healed specimen. This image affirms that the capsules were broken, and the encapsulated material was successfully healed since its texture seems similar to the texture of the epoxy resin, which constitutes the composite.

## 4. Conclusions

In this research, UF capsules with a nanomodified healing agent with MWCNTs were incorporated in GFRPs, aiming to increase the healing efficiency and the concurrent restoration of the electrical and mechanical properties. At mode-II tests, the introduction of the self-healing system improved the initial mechanical properties in the fracture toughness by ca. +28% in the case of neat capsules. The mechanical healing efficiency was calculated ca. 180%. The electrical recovery was calculated at approximately 61%. However, in the nanomodified capsule system, the G_IIc_ increased by +29%, after the incorporation of the healing system. The mode-II fracture toughness was regained ca. 191% while it exhibited an electrical recovery of ca. 79%. It is evident that with the nanomodification of the healing agent, the mechanical healing efficiency was significantly improved by ca. 10% and the electrical by 18%.

At the lap strap geometry or the modelled structure level, the incorporation of both neat and nanomodified capsules at the adhesive area had a negligible effect on the strength, showing that the employment of capsule-based methodologies did not affect the initial properties. The mechanical healing efficiency calculations exhibited a recovery of ca. 91% of the lap strap strength using neat capsules and ca. 107% recovery in the case of nanomodified capsules. A significant difference in the electrical recovery between the neat and nanomodified healing agent was more noticeable in this case. The DC conductivity recovery was increased from approximately 24% to 76%.

The calculated high mechanical and electrical healing efficiency values, along with the elimination of the knock-down effect, are promising factors for the scale-up of the employment of a nanomodified capsule-based self-healing method to model composite structures. Capsules with a nanomodified healing agent proved to be a promising healing method, especially for the application in self-sensing composites. They could restore the sensing functionality with the regeneration of the CNT network, that acts as a sensing element, even after a total disruption.

## Figures and Tables

**Figure 1 polymers-14-05264-f001:**
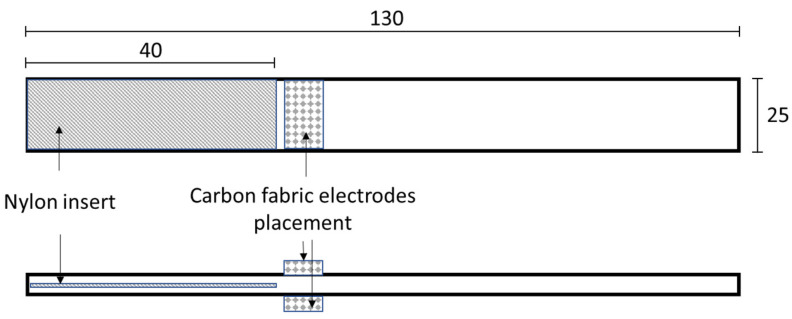
Schematic illustration of the attached carbon fabric electrodes setup on mode-II geometry and the dimensions in mm.

**Figure 2 polymers-14-05264-f002:**
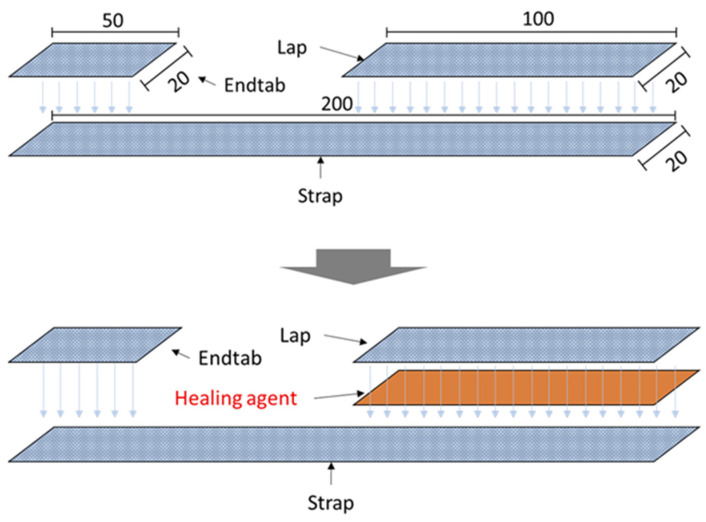
Lap strap geometry and the dimensions in mm (**up**) and measured self-healing section (**down**).

**Figure 3 polymers-14-05264-f003:**
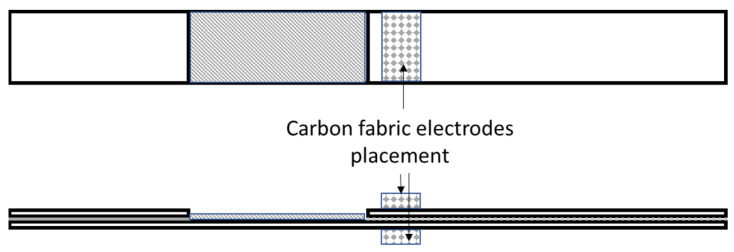
Schematic illustration of the attached carbon fabric electrodes setup on lap strap geometry.

**Figure 4 polymers-14-05264-f004:**
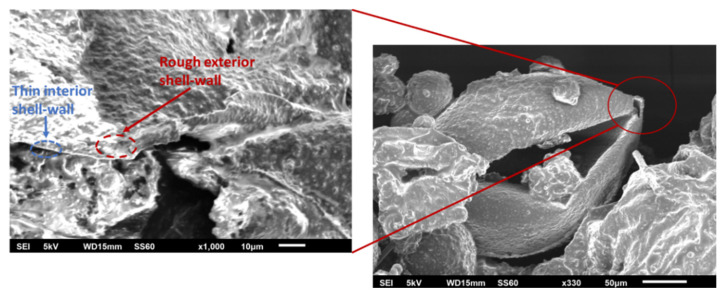
SEM images from capsules confirm the rough exterior and thin interior shell wall.

**Figure 5 polymers-14-05264-f005:**
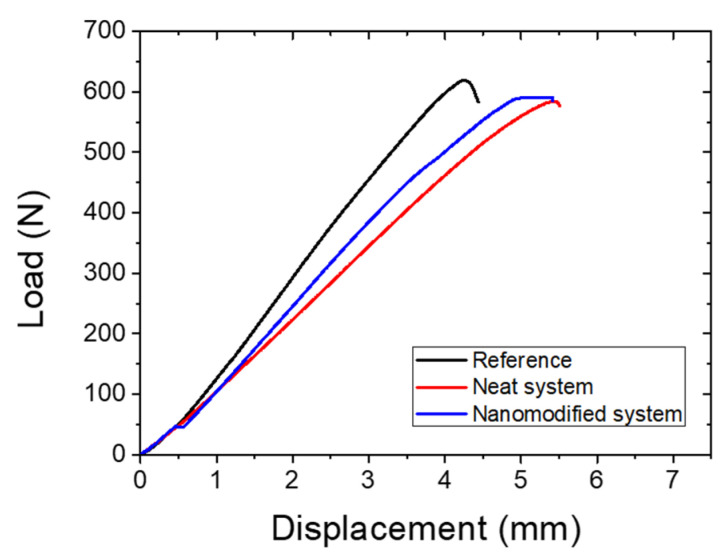
Representative plots of reference (black), neat (red line) and nanomodified (blue line) systems of mode-II specimens.

**Figure 6 polymers-14-05264-f006:**
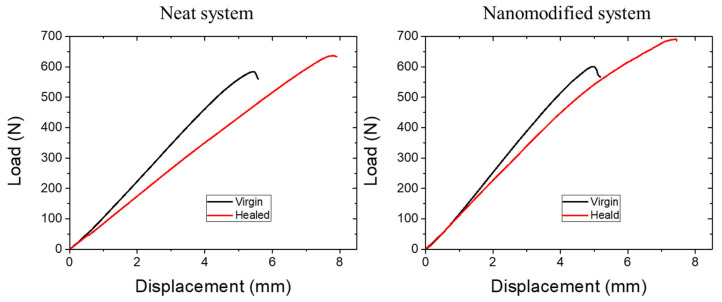
Representative plots of initial (black line) and healed (red line) mode-II specimens containing neat (**left**) and nanomodified (**right**) capsules.

**Figure 7 polymers-14-05264-f007:**
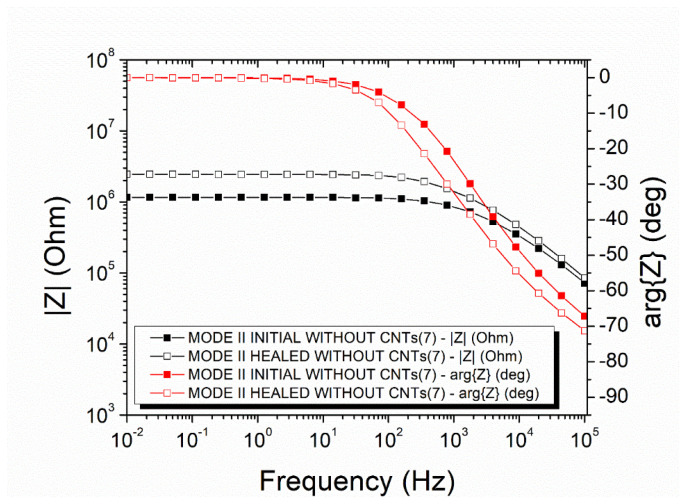
Representative impedance plots of a mode-II specimen with neat micro-capsules.

**Figure 8 polymers-14-05264-f008:**
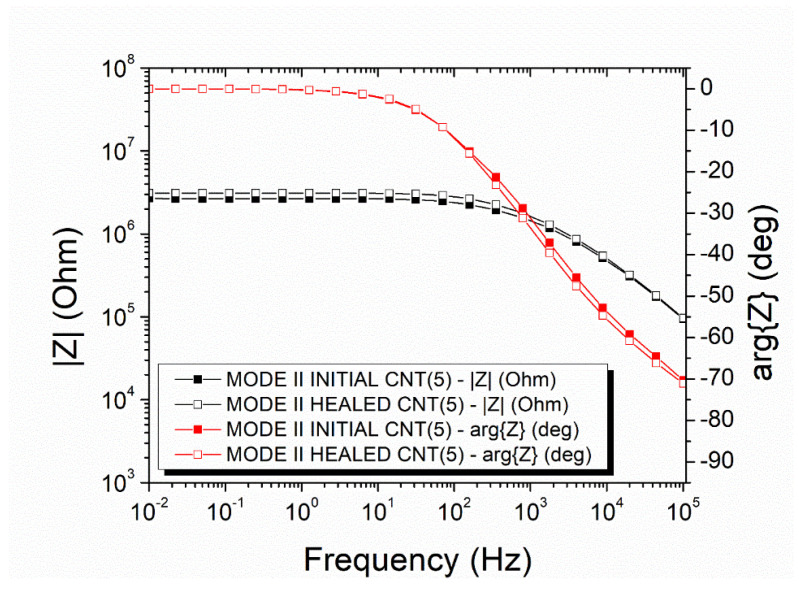
Representative Impedance plots of a mode-II specimen with nano-enhanced micro-capsules.

**Figure 9 polymers-14-05264-f009:**
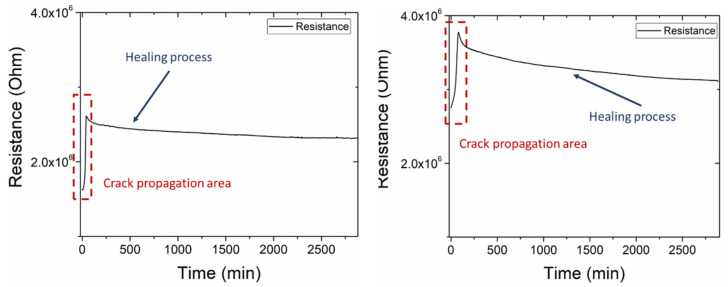
Online monitoring of resistance during the experimental procedure (damage and self-healing) of mode-II specimens containing neat (**left**) capsules and with a nano-enhanced healing agent (**right**).

**Figure 10 polymers-14-05264-f010:**
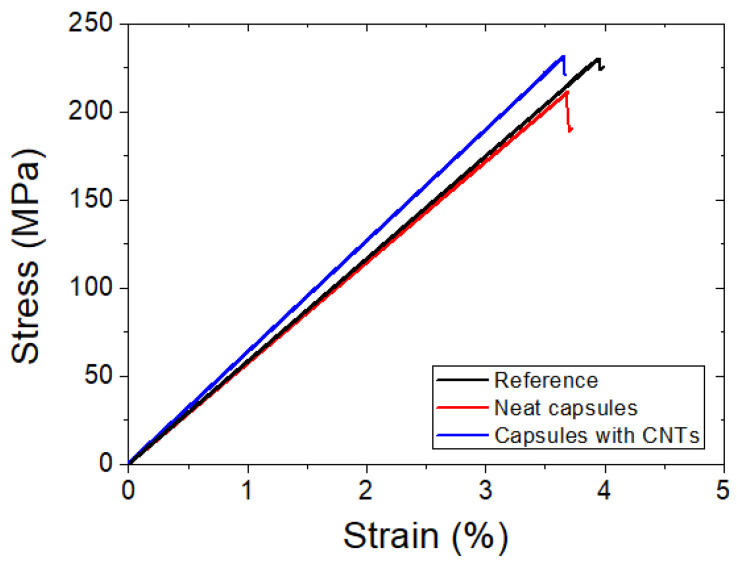
Representative plots of reference (black line), neat system (red line) and nanomodified system (blue line) of lap strap specimens.

**Figure 11 polymers-14-05264-f011:**
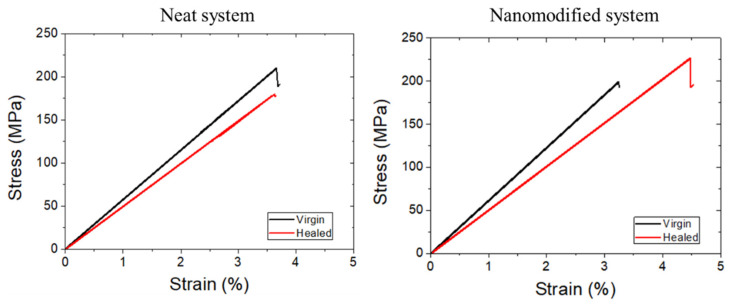
Representative plots of the virgin (black line) and healed (red line) lap strap specimens containing neat and nanomodified capsules.

**Figure 12 polymers-14-05264-f012:**
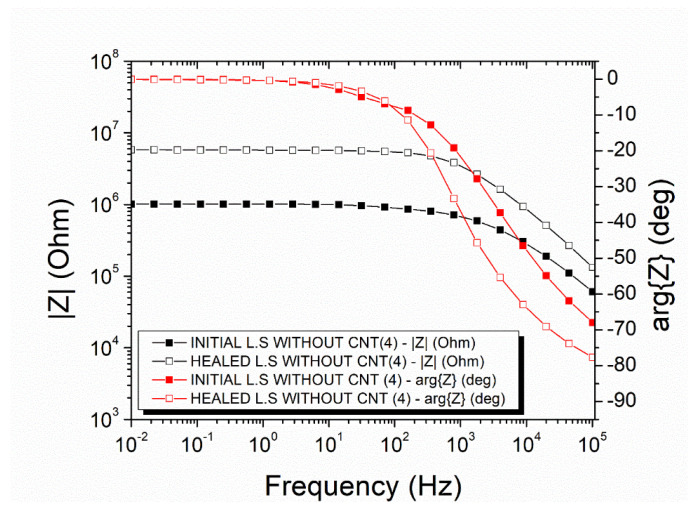
Impedance plots of lap strap specimens with neat micro-capsules.

**Figure 13 polymers-14-05264-f013:**
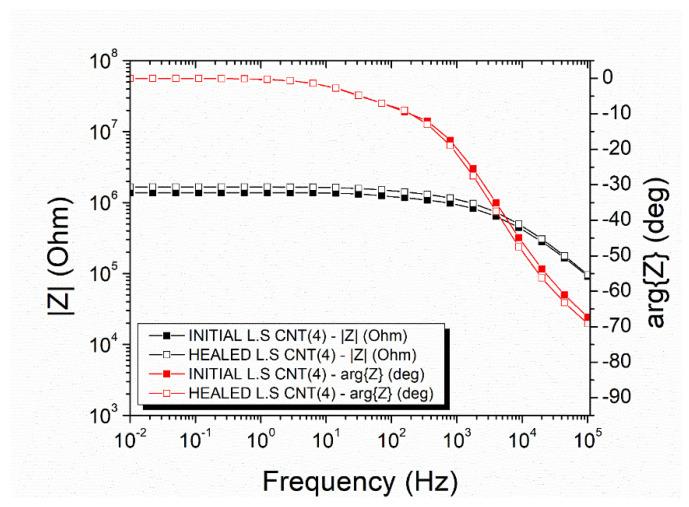
Impedance plots of lap strap specimens with nano-enhanced micro-capsules.

**Figure 14 polymers-14-05264-f014:**
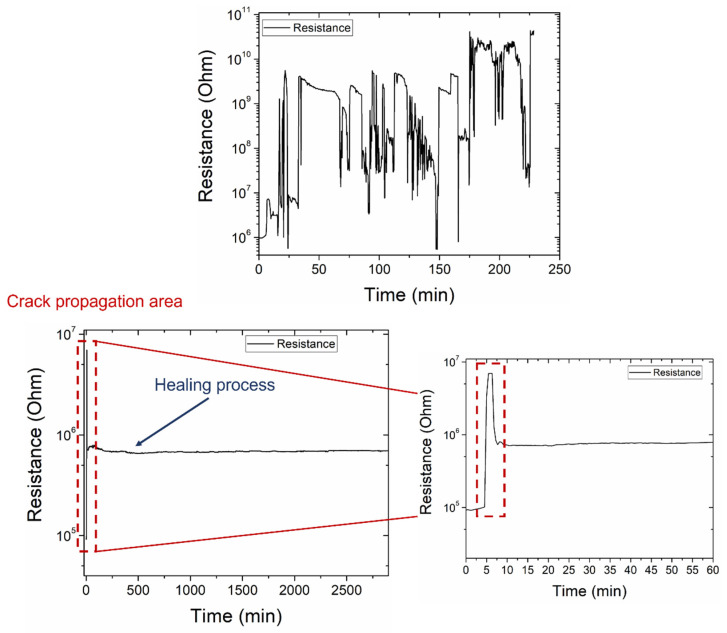
Online monitoring of resistance during the experimental procedure (damage and self-healing) of lap strap specimens containing neat (**up**) and capsules with nanomodified healing agent (**down**).

**Figure 15 polymers-14-05264-f015:**
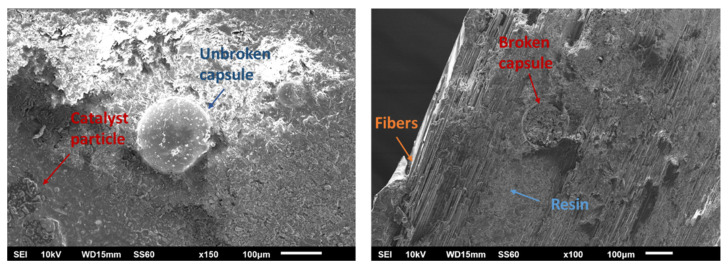
SEM images from mode-II specimens before (**left**) and after (**right**) healing.

**Table 1 polymers-14-05264-t001:** Healing efficiency and change in initial mechanical properties of mode-II specimens containing neat and nanomodified capsules in terms of G_IIc_.

System	Mean Diameter (μm)	G_IIc_ (kJ/m^2^) Initial	G_IIc_ (kJ/m^2^) Healed	Healing Efficiency (%)	Change in Initial Properties (%)
Reference		1.68 ± 0.11			
Neat capsules	205.81 ± 26.32	2.15 ± 0.32	3.89 ± 0.69	180.93	+27.98
Nanomodified capsules	167.25 ± 10.24	2.17 ± 0.35	4.15 ± 0.73	191.24	+29.17

**Table 2 polymers-14-05264-t002:** Healing efficiency and change in initial mechanical properties of mode-II specimens containing neat and nanomodified capsules in terms of maximum load.

System	Mean Diameter (μm)	Max Load (N) Virgin	Max Load (N) Healed	Healing Efficiency (%)	Change in Initial Properties (%)
Reference		616.00 ± 33.61			
Neat capsules	205.81 ± 26.32	574.14 ± 27.25	628.27 ± 28.41	109.42	−6.80
Nanomodified capsules	167.25 ± 10.24	598.57 ± 25.08	702.00 ± 42.29	117.28	−2.83

**Table 3 polymers-14-05264-t003:** Healing efficiency and change in initial mechanical properties of lap strap specimens containing neat and nanomodified capsules.

System	Mean Diameter (μm)	Max Stress (MPa) Virgin	Max Stress (MPa) Healed	Healing Efficiency (%)	Change in Initial Properties (%)
Reference		232.91 ± 7.33			
Neat capsules	205.81 ± 26.32	228.08 ± 17.24	207.33 ± 39.53	90.67	−2.07
Nanomodified capsules	167.25 ± 10.24	233.68 ± 18.25	240.07 ± 32.21	107.36	+0.03

## Data Availability

Not applicable.

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
