# Peer review of "Capsule-Based Self-Healing and Self-Sensing Composites with Enhanced Mechanical and Electrical Restoration"

_polymers, 2022, doi:10.3390/polym14235264_

Round 1

Reviewer 1 Report

1. 'Reference source not found': authors should double-check the manuscript to avoid it.

2. Significant figures are improperly chosen, such as thickness, load, efficiency, etc.

3. Why 'stress-strain' and 'load-displacement' were used at the same manuscript? Why not choose one?

4. More SEM observation should be added. For example, the samples before and after healing, 

5. Conclusion is too long to read, and please simplify it.

Author Response

Reviewer’s comment: 1. 'Reference source not found': authors should double-check the manuscript to avoid it.

Author’s response: Thank you for your comment. Unfortunately, there was an error in the word file after submitted to the journal. All the mistakes are corrected.

Reviewer’s comment: 2. Significant figures are improperly chosen, such as thickness, load, efficiency, etc.

Author’s response: Thank you for your comment. The figures were not mentioned in the manuscript due to the above error. All figures are mentioned in the manuscript.

Reviewer’s comment: 3. Why 'stress-strain' and 'load-displacement' were used at the same manuscript? Why not choose one?

Author’s response: The authors thank the reviewer for the comment. Load-displacement plots are presented in the case of Mode-II specimens. In addition, stress-strain curves are presented for the Lap Strap specimens. Since the lap-strap specimens are tested under tensile mode, stress-strain curves and results were calculated and presented as at typical tensile tests.

Reviewer’s comment: 4. More SEM observation should be added. For example, the samples before and after healing,

Author’s response: The authors thank the reviewer for the comment. The SEM cannot be performed on the samples before and after healing since the specimens would be cut and destroyed for the SEM observation.

Reviewer’s comment: 5. Conclusion is too long to read, and please simplify it.

Author’s response: The authors thank the reviewer for the comment. The conclusion is simplified.

Reviewer 2 Report

The manuscript having ID: polymers-1980347 which I reviewed discusses the manufacturing and characterization of self-healing and self-sensing nano-composites. Impedance Spectroscopy and Mode II testing methods were adopted for performance evaluation.

The contribution of the manuscript is worthy to be published. I recommend to publish this manuscript with the following modifications/responses:

1.               The language shall be improved. Remove ‘too’ in the abstract in line 11.

2.               Always separate the unit with a space from its magnitude (5-10 μm). Correct throughout the manuscript.

3.               Refer all figures correctly in the text.

4.               In figure 1, show the dimensions on the figure for readily reference to the readers.

5.               Mention the applicable ASTM standard followed for Mode II.

6.               Enhance the presentation of mathematical equations.

7.               Include equipment details for SEM. Enhance figure 4.

8.               How the filtration of the nanoparticles during composite fabrication was overcome? Cite the following articles which also discuss this problem and the fabric was coated rather than mixing the nanomaterials in the epoxy.

https://doi.org/10.1016/j.compstruct.2022.116169

                 https://doi.org/10.1016/j.coco.2022.101382

Author Response

Reviewer’s comment: The manuscript having ID: polymers-1980347 which I reviewed discusses the manufacturing and characterization of self-healing and self-sensing nano-composites. Impedance Spectroscopy and Mode II testing methods were adopted for performance evaluation. The contribution of the manuscript is worthy to be published. I recommend to publish this manuscript with the following modifications/responses:

Author’s response: The authors thank the reviewer for the comment.

Reviewer’s comment: 1. The language shall be improved. Remove ‘too’ in the abstract in line 11.

Author’s response: Thank you for your comment. The “too” was removed and the manuscript improved.

Reviewer’s comment: 2. Always separate the unit with a space from its magnitude (5-10 μm). Correct throughout the manuscript.

Author’s response: Thank you for your comment. The manuscript was corrected.

Reviewer’s comment: 3. Refer all figures correctly in the text.

Author’s response: The authors thank the reviewer for the comment. Unfortunately, there was an error in the word file after submitted to the journal. All mistakes are corrected, and the figures are referred correctly.

Reviewer’s comment: 4. In figure 1, show the dimensions on the figure for readily reference to the readers.

Author’s response: The authors thank the reviewer for the comment. The dimensions are added to the figure.

Reviewer’s comment: 5. Mention the applicable ASTM standard followed for Mode II.

Author’s response: The authors thank the reviewer for the comment. The Mode-II specimens were manufactured and tested according to the AITM 1.0006 (Airbus Industrie Test Method) and not by ASTM. This was mentioned in lines 147 and 193.

Reviewer’s comment: 6. Enhance the presentation of mathematical equations.

Author’s response: Thank you for your comment. All equations are corrected.

Reviewer’s comment: 7. Include equipment details for SEM. Enhance figure 4.

Author’s response: Thank you for your comment. Details for the SEM instrument were added.

Reviewer’s comment: 8. How the filtration of the nanoparticles during composite fabrication was overcome? Cite the following articles which also discuss this problem and the fabric was coated rather than mixing the nanomaterials in the epoxy. https://doi.org/10.1016/j.compstruct.2022.116169

https://doi.org/10.1016/j.coco.2022.101382

Author’s response: The authors thank the reviewer for the comment. The nanoparticles were dispersed into the epoxy resin that was used as matrix via high-shear mixing and were not coated at the fibers. With this process, there were not any excess CNTs or precipitate in the resin. The method/articles that the reviewer proposes referred to the sensing technology with coated fibers that is based on other principles and the sensing elements were the fibers. In this study, the sensing element is the nanomodified matrix or the conductive network that is capable of detecting changes in the structure of the composite with high sensitivity. However, we have cited the papers as they are relevant to this study.

Round 2

Reviewer 1 Report

Not all the comments were carefully responsed.

1. Significant digits or significant figures are improperly chosen, such as thickness, load, efficiency, etc. 

2. Why 'stress-strain' and 'load-displacement' were used at the same manuscript? Why not choose one? Or why not using the same type of samples in this work?

3. More SEM observation must be added. For example, the samples before and after healing. It is possible to carry out the SEM observation because many works reported the morphologies of the samples before and after healing.

Author Response

1st reviewer

Reviewer’s comment: 1. Significant digits or significant figures are improperly chosen, such as thickness, load, efficiency, etc.

Author’s response: Thank you for your comment. All significant digits were corrected.

Reviewer’s comment: 2. Why 'stress-strain' and 'load-displacement' were used at the same manuscript? Why not choose one? Or why not using the same type of samples in this work?

Author’s response: The authors thank the reviewer for the comment. Load-displacement plots are presented in the case of Mode-II specimens. In this case according to the standard (AITM-1-0006), the load applied to the specimen and the cross-head displacement of the test machine are recorded continuously during the test, thus we presented the load-displacement curves. The following sentence was added in lines 197-198:

“The displacement rate was set at 1 mm/min while load and cross-head displacement, were recorded.”

In addition, stress-strain curves are presented for the Lap Strap specimens. According to the CWA 17896:2022 “Test method for the evaluation of the adhesive properties of fibre reinforced polymer composite joints”, the stress and the strain are recorded continuously during the test, thus we presented the stress-strain curves. Also, some representative references of similar works in this geometry are 17 and 42 from the manuscript. The following sentence was added in line 207:

“Stress and strain were recorded continuously during the test.”

Moreover, in this work, the Mode-II test was used in order to measure the Mode-II fracture toughness properties. With the lap strap test, the bonding properties of the adhesive layer between the two parts were measured.

Reviewer’s comment: 3. More SEM observation must be added. For example, the samples before and after healing. It is possible to carry out the SEM observation because many works reported the morphologies of the samples before and after healing.

Author’s response: Thank you for your comment. As the reviewer correctly pointed out, the SEM observation was not added to the manuscript as the morphology of the samples has already been reported. In addition, the samples for SEM observation have to be cut as well as destroyed, so it was not possible to perform the study. However, we added SEM figures from Mode-II specimens that were excluded from the calculation of the results and manufactured only for the fractography.

Reviewer 2 Report

All the comments have been addressed in the revised manuscript.

Author Response

Reviewer’s comment: 1. Significant digits or significant figures are improperly chosen, such as thickness, load, efficiency, etc.

Author’s response: Thank you for your comment. All significant digits were corrected.

Reviewer’s comment: 2. Why 'stress-strain' and 'load-displacement' were used at the same manuscript? Why not choose one? Or why not using the same type of samples in this work?

Author’s response: The authors thank the reviewer for the comment. Load-displacement plots are presented in the case of Mode-II specimens. In this case according to the standard (AITM-1-0006), the load applied to the specimen and the cross-head displacement of the test machine are recorded continuously during the test, thus we presented the load-displacement curves. The following sentence was added in lines 197-198:

“The displacement rate was set at 1 mm/min while load and cross-head displacement, were recorded.”

In addition, stress-strain curves are presented for the Lap Strap specimens. According to the CWA 17896:2022 “Test method for the evaluation of the adhesive properties of fibre reinforced polymer composite joints”, the stress and the strain are recorded continuously during the test, thus we presented the stress-strain curves. Also, some representative references of similar works in this geometry are 17 and 42 from the manuscript. The following sentence was added in line 207:

“Stress and strain were recorded continuously during the test.”

Moreover, in this work, the Mode-II test was used in order to measure the Mode-II fracture toughness properties. With the lap strap test, the bonding properties of the adhesive layer between the two parts were measured.

Reviewer’s comment: 3. More SEM observation must be added. For example, the samples before and after healing. It is possible to carry out the SEM observation because many works reported the morphologies of the samples before and after healing.

Author’s response: Thank you for your comment. As the reviewer correctly pointed out, the SEM observation was not added to the manuscript as the morphology of the samples has already been reported. In addition, the samples for SEM observation have to be cut as well as destroyed, so it was not possible to perform the study. However, we added SEM figures from Mode-II specimens that were excluded from the calculation of the results and manufactured only for the fractography.

Round 3

Reviewer 1 Report

This manuscript can be accepted in present form.